# Structure of the plastic-degrading *Ideonella sakaiensis* MHETase bound to a substrate

Gottfried J. Palm [1], Lukas Reisky [2], Dominique Böttcher[2], Henrik Müller[2], Emil A.P. Michels[1], Miriam C. Walczak[2], Leona Berndt[1], Manfred S. Weiss[3], Uwe T. Bornscheuer [2] & Gert Weber [1,4]

The extreme durability of polyethylene terephthalate (PET) debris has rendered it a long-term environmental burden. At the same time, current recycling efforts still lack sustainability. Two recently discovered bacterial enzymes that specifically degrade PET represent a promising solution. First, *Ideonella sakaiensis* PETase, a structurally well-characterized consensus α/β-hydrolase fold enzyme, converts PET to mono-(2-hydroxyethyl) terephthalate (MHET). MHETase, the second key enzyme, hydrolyzes MHET to the PET educts terephthalate and ethylene glycol. Here, we report the crystal structures of active ligand-free MHETase and MHETase bound to a nonhydrolyzable MHET analog. MHETase, which is reminiscent of feruloyl esterases, possesses a classic α/β-hydrolase domain and a lid domain conferring substrate specificity. In the light of structure-based mapping of the active site, activity assays, mutagenesis studies and a first structure-guided alteration of substrate specificity towards bis-(2-hydroxyethyl) terephthalate (BHET) reported here, we anticipate MHETase to be a valuable resource to further advance enzymatic plastic degradation.

[1] Molecular Structural Biology, University of Greifswald, Felix-Hausdorff-Str. 4, 17487 Greifswald, Germany. [2] Biotechnology & Enzyme Catalysis, University of Greifswald, Felix-Hausdorff-Str. 4, 17487 Greifswald, Germany. [3] Macromolecular Crystallography, Helmholtz-Zentrum Berlin für Materialien und Energie, Albert-Einstein-Straße15, 12489 Berlin, Germany. [4] Present address: Macromolecular Crystallography, Helmholtz-Zentrum Berlin für Materialien und Energie, Albert-Einstein-Straße 15, 12489 Berlin, Germany. These authors contributed equally: Gottfried J. Palm, Lukas Reisky. Correspondence and requests for materials should be addressed to U.T.B. (email: uwe.bornscheuer@uni-greifswald.de) or to G.W. (email: gert.weber@helmholtz-berlin.de)

Appreciating its simple synthesis, robustness and durability, industrial production of PET was launched soon after its discovery and has been gradually increasing, projected to be over 70 million tons in 2020[1,2]. One of the biggest advantages of PET is its chemical inertness due to the hydrophobicity of the terephthalic acid (TPA) moiety, rendering it nearly resistant to environmental degradation. Although PET and other synthetic polymer plastics are considered nontoxic, larger particles and micro granules thereof are durable, omnipresent in marine or terrestrial habitats and accumulate in living organisms[3–5]. Often, they are also the carriers of potentially toxic colorants and additives[5–7]. Current recycling efforts cover only a fraction of PET waste and yield downgraded lower value products. They depend on the addition of large quantities of virgin polymer and significant consumption of energy[4]. Alternatively, several enzymes have been identified that can hydrolyze PET to TPA and ethylene glycol at elevated temperatures, albeit with low activity[8–11]. Enzyme optimization by biotechnology has been successful to some degree[12–17], but has so far not led to enzymes, which can fully penetrate and degrade a thick layer of highly crystalline PET in a cost-effective and environmentally friendly manner.

Recently, the bacterial strain *Ideonella sakaiensis* 201-F6 was discovered and shown to grow on low-crystallinity PET films. Two α/β-hydrolase fold enzymes (α/β-hydrolases), PETase and MHETase, work together to degrade PET in two steps via MHET, yielding TPA and ethylene glycol—the building blocks required for a new round of PET synthesis (Fig. 1a)[10,18]. Recent crystal structures of PETase bound to ligands confirmed the predicted α/β-hydrolase fold, elucidated substrate binding, mode of catalysis and even permitted the enhancement of catalytic properties or alteration of substrate specificity[16,17,19–21]. Compared to known

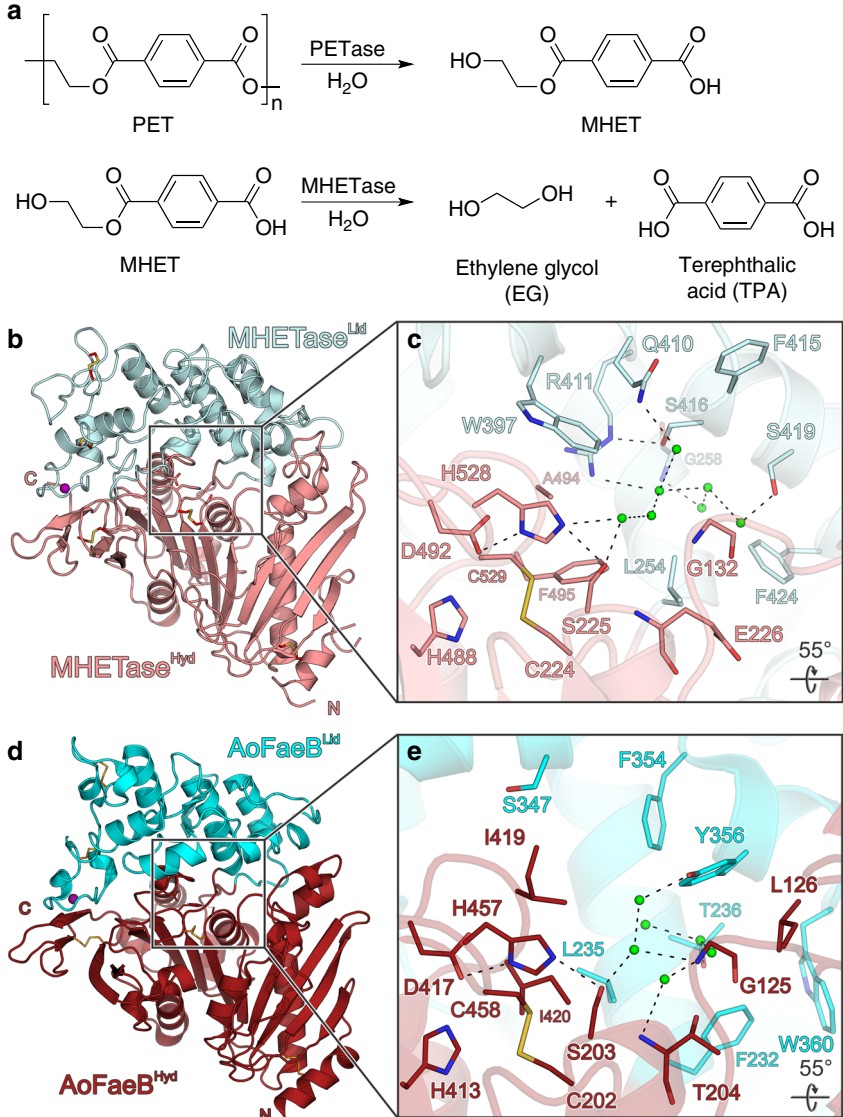

**Fig. 1** The structure of *I. sakaiensis* MHETase displays a bipartite domain architecture. **a** *I. sakaiensis* PETase and MHETase degrade PET to terephthalic acid and ethylene glycol. Side products are not shown. **b** MHETase structure with the α/β-hydrolase domain (MHETase^Hyd) colored in salmon and the lid domain (MHETase^lid) in light blue. Disulfide bonds are shown as sticks. **c** Close-up view of the MHETase catalytic triad, oxyanion hole and the water molecules in the substrate-binding site. **d** *A. oryzae* FaeB (PDB-ID: 3WMT[24]), α/β-hydrolase domain (AoFaeB^Hyd) in crimson red, lid domain (AoFaeB^Lid) in cyan. **e** Close-up view of the AoFaeB catalytic triad, oxyanion hole and the water molecules in the substrate-binding site. Dashed lines indicate hydrogen bonds, rotation angles relate to the overview. Interacting residues are shown as sticks and colored by atom type. Carbon—as given for the respective molecule; nitrogen—blue; oxygen—red; sulfur—yellow. Water oxygens are shown as green spheres. Calcium is shown as purple sphere

PET-degrading esterases, PETase from *I. sakaiensis* shows higher activity at ambient temperature and on highly crystalline PET[10]. In contrast, the structure of *I. sakaiensis* MHETase, the second enzyme—and crucial for full PET degradation—is still unknown. MHETase was initially assigned to the tannase enzyme family, which belongs to Block X of the α/β-hydrolase fold enzymes classified in the ESTHER database[10,22]. This family includes fungal and bacterial tannases and feruloyl esterases. Other significantly different bacterial tannases can be found in a distinct Block H (Tannases_bact) in this database. Consistently, MHETase was shown to exclusively hydrolyze MHET but not BHET, PET, *p*-nitrophenyl (pNP) aliphatic esters or aromatic ester compounds such as ethyl gallate and ethyl ferulate which are converted by other enzymes from the tannase family, indicating a highly restricted substrate specificity[10]. All plastic-degrading enzymes known so far display an α/β-hydrolase fold. MHETase, however, is likely to possess a scaffold unprecedented for plastic-degrading enzymes. This may be exploited in order to improve catalysis and to expand substrate specificity and thus significantly advance enzymatic plastic polymer degradation.

Here, we present the crystal structures of *I. sakaiensis* PETase, MHETase and MHETase bound to a nonhydrolyzable substrate analog (MHETA) or to benzoic acid. A structure-based mapping of the active site by mutations and binding studies with different substrates was used to determine the molecular basis for product inhibition and guided the development of MHETase variants with enhanced activity towards MHET or even an altered substrate specificity towards BHET. We anticipate our data to significantly advance the current understanding of enzymes degrading synthetic polyesters.

## Results

**Structure and phylogeny of *I. sakaiensis* MHETase**. We have determined crystal structures of recombinantly expressed and purified *I. sakaiensis* MHETase in its ligand-free form (2.05 Å resolution), MHETase bound to a nonhydrolyzable mono-(2-hydroxyethyl) terephthalamide (MHETA, 2.1 Å resolution) or to benzoic acid (BA, 2.2 Å resolution) as well as ligand-free PETase (2.0 Å resolution) (Supplementary Figs. 1, 2a−f, Supplementary Table 1). The structure of PETase was solved by molecular replacement (MR) employing the structural coordinates of *T. fusca* cutinase TfCut2 (PDB entry 4CG1[11,23]; see Methods). The structure of MHETase was solved by an MR pipeline employing a recent feruloyl esterase structure (PDB entry 6G21; see Methods). The overall domain architecture of the 65 kDa MHETase resembles that of feruloyl esterases, with a lid domain inserted between β-strand 7 and α-helix 15 of the α/β-hydrolase fold (Fig. 1b, Supplementary Fig. 1).

As previously observed for feruloyl esterases, the presence of a structural calcium-binding site was confirmed by X-ray fluorescence spectroscopy for MHETase (Fig. 1b, Supplementary Fig. 2b). Likewise, one of five disulfide bonds is flanking a catalytic triad (formed by S225, H528, D492) and the oxyanion hole comprising the backbone amide nitrogen atoms of G132 and E226 (Fig. 1c)[24]. In the ligand-free structure of MHETase, several water molecules are maintained by a hydrogen bond network at the substrate-binding site (Fig. 1c). While the α/β-hydrolase domain superimposes well with the closest structurally characterized feruloyl esterase homolog FaeB from *A. oryzae* (1.60 Å RMSD for 280 out of 342 residues aligned, 32.5% amino acid identity), the lid domain of MHETase contains several additional loops that markedly differ from FaeB (2.33 Å RMSD for 148 out of 215 residues aligned, 18.9% identity) (Fig. 1b, d)[24]. The overall structures of MHETase and FaeB are structurally similar (2.04 Å RMSD for 421 out of 559 residues aligned) despite a relatively low

number of amino acid identities (27.5%). When comparing MHETase with known tannase structures, e.g tannin acyl α/β-hydrolase from *Lactobacillus plantarum* (LptE), it is evident that only the overall fold of the α/β-hydrolase domain is similar (2.77 Å RMSD for 195 out of 282 residues aligned, 13.8% identity), while very large differences (5.24 Å RMSD) are observed for the lid domain (Supplementary Fig. 3a, b)[25]. PETase and MHETase only share the α/β-hydrolase fold (2.87 Å RMSD for 184 out of 262 residues aligned; Supplementary Fig. 3c).

A phylogenetic analysis groups MHETase with the feruloyl esterases and tannases of Block X in the ESTHER database. It is located in a branch with no other structures (Supplementary Fig. 4). The structures of the closest MHETase relatives are 3WMT and 6G21, two feruloyl esterases of *Aspergillus oryzae*. With them, MHETase shares not only the catalytic triad S225-H528-D492 (3WMT: S203-H457-D417, 6G21: S169-H421-D381) but also G132 (G125, G91) as part of the oxyanion hole and C224-C529 (C202-C528, C168-C422), whose disulfide bond holds the catalytic residues Ser and His together[24]. All these residues are in the catalytic domain, the disulfide bond is typical (>80% conservation) for the tannase family in Block X of α/β-hydrolases according to the ESTHER database (Fig. 1b–e)[22,24]. The lid domain of the feruloyl esterases displays the same α-helical fold, but the amino acid sequences cannot be aligned without structural information. A binding pocket—like that for MHET in MHETase—also exists in feruloyl esterases, but not a single one of the lining residues is conserved. However, a comparison of the MHETase active site to that of FaeB reveals several residues around the catalytic triad that may contribute to substrate positioning in a similar fashion (e.g. L235, F354 and L245, F415 in FaeB and MHETase, respectively) (Fig. 1c, e). Most likely, it is the alteration in the substrates, especially the carboxylic acid group of MHET vs. the phenolic (methyl ether) groups and the elongating double bond of ferulates that has evoked this difference. The substrates of tannases, e.g. gallates, are more similar at least with respect to their size to MHET. The only tannase structure available in complex with ethyl gallate is from *Lactobacillus plantarum* (4J0K), which belongs to the bacterial tannase family in Block H of α/β-hydrolases according to the ESTHER database[22,25]. The tannase catalytic domains are sufficiently conserved, such that the catalytic triad superimposes well. However, the sequences differ strongly, the disulfide bond is missing and the lid domain has a markedly different fold.

**Structure of MHETase bound to a nonhydrolyzable ligand**. The main chain conformation in the MHETase-MHETA complex structure is nearly identical to that of MHETase without substrate (RMSD 0.54 Å) and sheds light on the positioning of MHET for catalysis. While the catalytic triad and oxyanion hole residues are part of the α/β-hydrolase domain, substrate specificity is almost exclusively conferred by the lid domain (Fig. 2a, b). Hydrophobic contacts between the phenyl ring of MHETA and the α/β-hydrolase domain are restricted to primarily F495, and to a lesser extent G132 and A494. Strikingly, MHETA is tightly bound by the lid domain residues F415, L254 and W397 surrounding nearly the entire MHETA phenyl moiety. The two oxygens of the free carboxylate make contacts to R411, which is held in place by S416, S419 and the backbone amide of G258, which maintain a hydrogen bond network involving three water molecules.

Despite their overall high similarity, a detailed comparison of MHETase structures in the absence and presence of the substrate reveals an induced-fit mechanism upon MHETA binding (Fig. 2c, d). In the ligand-free structure, F415 points away from

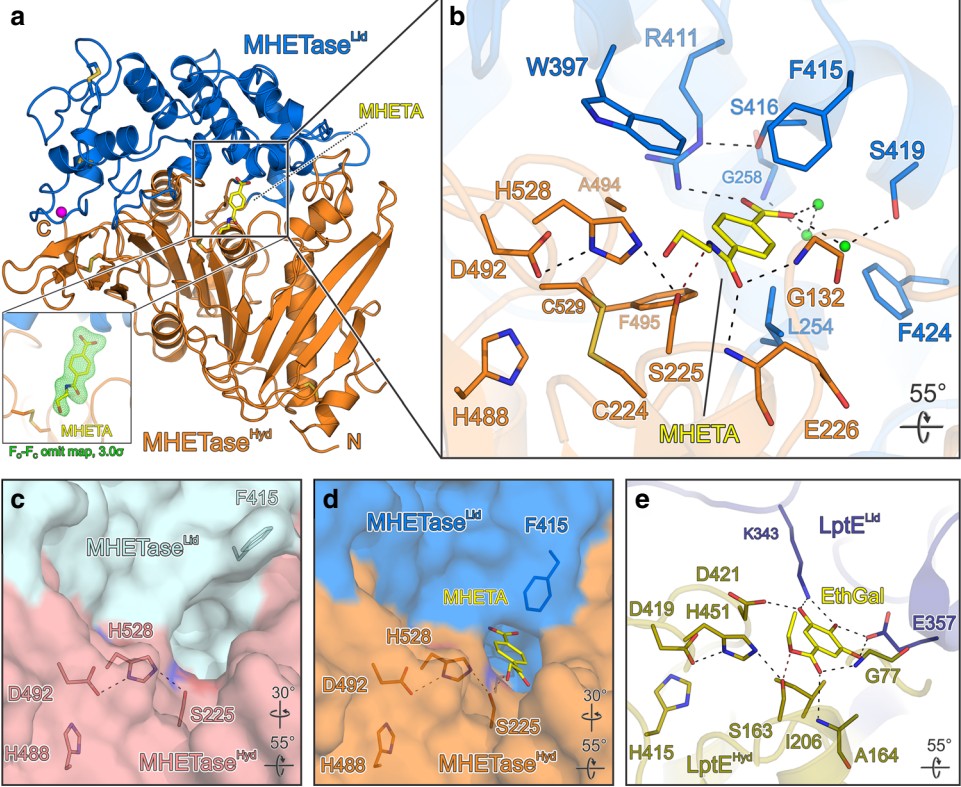

**Fig. 2** Structure of *l. sakaiensis* MHETase bound to a nonhydrolyzable MHET analog. The structure explains substrate specificity and reveals an induced-fit substrate-binding mode. **a** Co-structure of MHETase bound to MHETA (yellow), α/β-hydrolase domain (MHETase^Hyd) in orange, lid domain (MHETase^Lid) in marine blue. Inset, bottom left—refined $F_O-F_C$-omit electron density map (green) contoured at 3σ for MHETA. MHETA of the refined final structure is shown as sticks. **b** Close-up view on MHETA (yellow) bound to the active site of MHETase. **c**, **d** Molecular surface of the MHETase active site **c** without and **d** with bound MHETA, the catalytic triad and F415 are shown as sticks. **e** Close-up view of the tannin acyl α/β-hydrolase active site from *L. plantarum* (PDB-ID: 4J0K[25]) bound to ethyl gallate (EthGal, yellow), superimposed on helix α5 of MHETase (not shown). α/β-Hydrolase domain (LptE^Hyd) in olive, lid domain (LptE^Lid) in dark blue. Rotation symbols indicate views relative to **a**. Color scheme for interacting residues and water oxygens as in Fig. 1. Calcium is shown as magenta spheres

the active site and thus opens it for substrate binding. The association of MHETA then triggers a near 180° rotation of the F415 side chain around χ₁, closing the active site and consolidating the interaction.

Lastly, unlike PETase, MHETase binds to its substrate very tightly with a $K_m$ of 7.3 μM[17]. A comparison of the active-site molecular surfaces of LptE, PETase and MHETase in their substrate-bound states illustrates a higher solvent accessibility of LptE and PETase, which is partially related to the induced-fit mechanism observed for MHETase and the number of residues contacting the respective substrate (Fig. 2c, d, Supplementary Figure 5 a–d).

The positioning of the substrate in the active site of MHETase is reminiscent of the tannin acyl α/β-hydrolase from *L. plantarum* bound to ethyl gallate (LptE) but displays marked differences with respect to the residues contacting the substrate at the interface (Fig. 2b, e). In the LptE structure, the phenyl moiety of ethyl gallate is exclusively contacted by I206 and G77 of the α/β-hydrolase domain while the three hydroxyl groups are hydrogen-bonded to D421 of the α/β-hydrolase domain and to K343 and E357 of the lid domain. Thus, the contribution of the LptE lid domain to substrate binding is much reduced, compared to the situation in MHETase.

**MHETase ligand spectrum and implications for the active site.** The substrate-binding position of PETase fundamentally differs

from that of MHETase as shown for the PETase-1-(2-hydroxyethyl) 4-methyl terephthalate (HEMT) and *p*-nitrophenol (pNP) co-structures (Supplementary Figure 5 a–c). In particular for PETase, the absence of a lid domain limits the number of residues involved in immediate substrate recognition down to four.

Primarily the phenyl moiety of HEMT and also pNP is bound by hydrophobic contacts of Y85, M132, W156 and I179, but the methyl ester or hydroxyl group in the 4-position of HEMT or pNP, respectively, is fully exposed to the bulk solvent (Supplementary Figure 5 a–c). In summary, the complexity of substrate recognition by MHETase clearly distinguishes it from other enzymes, such as tannases or even PETase.

Apart from the entire substrate, even substrate sub-structures and analogs such as benzoic acid or nicotinic acid are able to bind tightly to MHETase as observed in the respective co-structure and in differential scanning fluorimetry (DSF) measurements (Supplementary Figure 6a, Fig. 3). It is again mainly lid domain residues, which establish the contacts to benzoic acid and place it in an identical position as MHETA (Supplementary Figure 6b).

A comparison of potential MHETase ligands with limited variations by DSF confirms which functional groups are recognized by the MHETase binding site (for ligand quality control, see Supplementary Figure 7a–f). Our structural analysis suggests that R411 enforces a strictly required negative charge in the 4-position to the hydrolyzed ester bond, which clearly

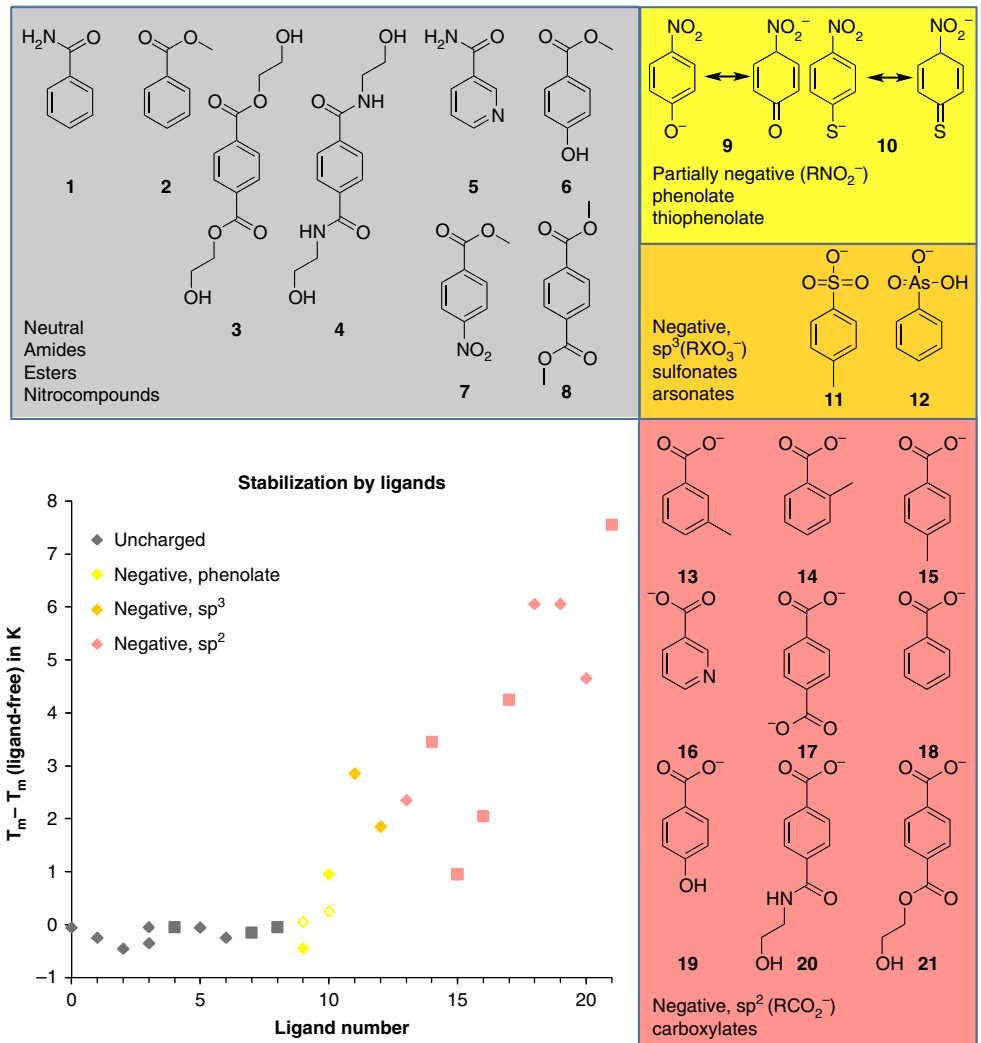

**Fig. 3** Differential scanning fluorimetry of MHETase with various ligands. Stabilization of MHETase wild type as measured by an increase in the melting temperature ($T_m$) compared to the ligand-free protein ($T_{m,\ ligand-free}$) is clearly dependent on the functional group, which binds to the rear of the substrate-binding site. Neutral functional groups (gray) affect $T_m$ weakest, partially negative groups (yellow) weakly, negatively charged groups more strongly, depending on their geometry: tetrahedrally coordinated medium (orange) and trigonal planar most strongly (red). Ligands diluted to 10 mM final concentration (full diamonds). A saturated solution (with 43.5% DMSO) was diluted 2.2-fold, if the compounds did not fully dissolve (squares). For compounds hindering reliable measurements by absorption (4-nitrophenol, 4-nitrothiophenol, 2-hydroxybenzoic acid) or fluorescence (BHET) 1 mM final concentrations were used (empty diamonds). $T_m$ values are reported as provided by the Prometheus software (maximum of the slope for the $I_{330\ nm}/I_{350\ nm}$ ratio). The experiment was performed as a single measurement. Ligand numbering is shown in Supplementary Table 3

explains why diesters and diamides show no binding to MHETase. BHET binding is thus excluded and MHET will only bind in the proper orientation but not with the hydroxyethyl group buried in the substrate pocket (Fig. 2b). Nitro groups can bind weakly when partially charged as a resonance structure with a phenolate or thiophenolate group. Negatively charged tetrahedral groups (sp³) as in sulfonic and arsenic acids can bind to MHETase but are clearly outperformed by the planar (sp²) group of the carboxylate (Fig. 3). Consequently, the loss of the positive charge in the MHETase R411Q and R411A variants leads to strongly reduced substrate binding and decreased inhibition by benzoate in mono-4-nitrophenyl terephthalate (MpNPT) hydrolysis (Supplementary Figure 8, Supplementary Table 2).

**Effect of MHETase mutants and generation of BHETase activity.** The central role of R411 in coordinating the carboxylic

acid function of the substrate was additionally confirmed by activity assays with the variants R411A and R411Q. These mutants show a strong increase of $K_m$ and some decrease in turnover rate against MpNPT (Fig. 4a). Furthermore, R411A and R411Q mutations almost completely abolish the conversion of the natural substrate MHET (Fig. 4b). If the hydrogen bond of the substrate carboxylate to S416 or S419 is also abolished in double mutants, $K_m$ further increases to about 1000-fold over the wild-type level (Fig. 4a). Thus, substrate recognition strongly relies on the aromatic ring as well as the carboxylate function of MHET both guiding its positioning for hydrolysis. Inhibitor experiments with benzoate derivatives and R411 and S416 mutants also demonstrate the importance of the interaction between the carboxylate moiety of the ligand and R411 together with hydrogen bonding for tight binding (Supplementary Table 2).

The high affinity for compounds with a benzoate substructure is expected to lead to product inhibition by formed TPA when

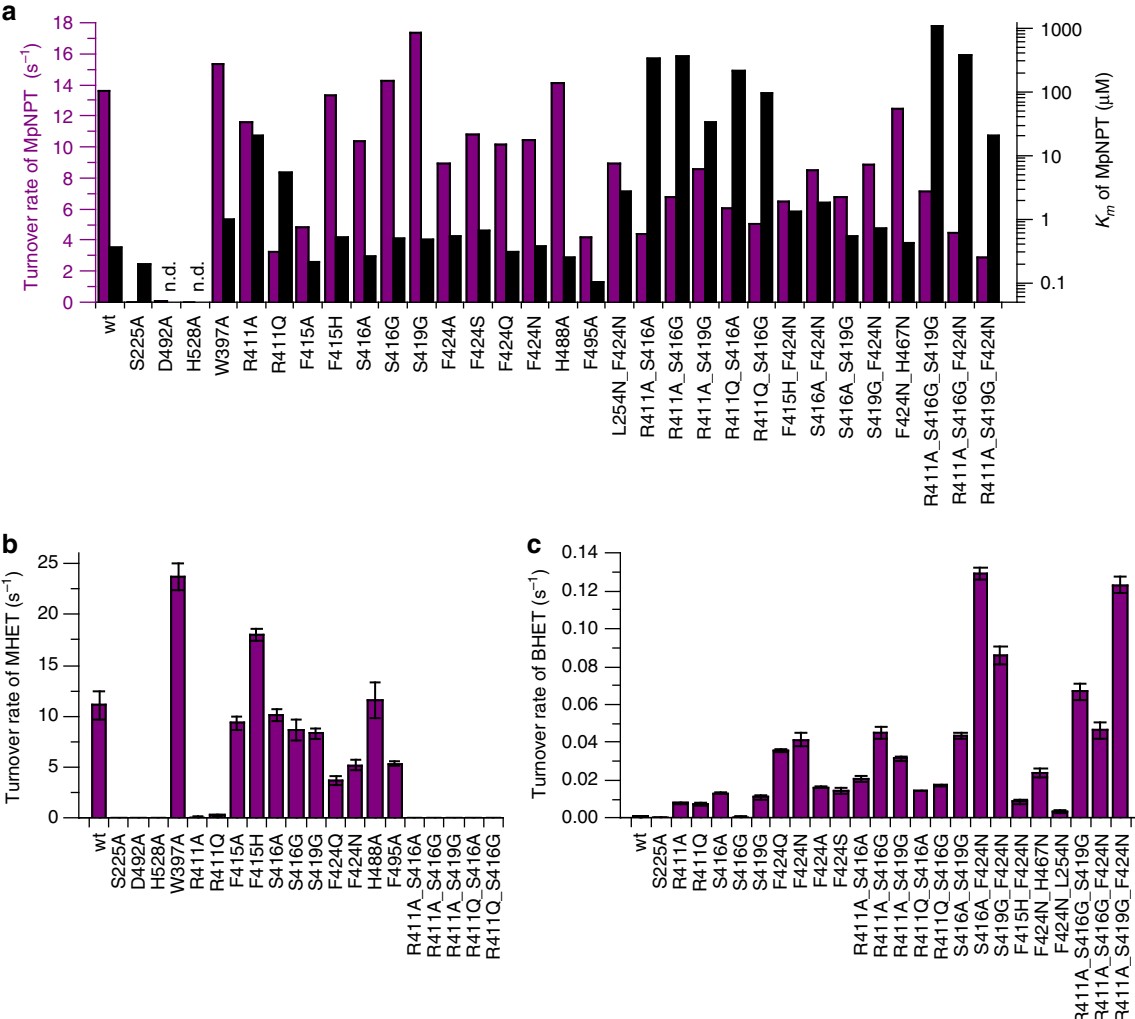

**Fig. 4** Catalytic properties of MHETase active-site mutants. **a** Kinetics of the conversion of MpNPT. The turnover rate (purple $y$-axis and bars) and $K_m$ (black $y$-axis and bars) were determined by activity measurements at varying substrate concentrations at 25 °C with the different active-site mutants ($x$-axis). See methods for the fitting procedure. **b** Turnover of MHET. **c** Turnover of BHET with selected mutants. The turnover rates in **b** and **c** were determined by quantification of hydrolysis products via HPLC analysis and represents the mean value. Error bars indicate the standard deviation of triplicates in these measurements. n.d.: $K_m$ for these active-site mutants could not be determined

higher concentrations of MHET are hydrolyzed in vitro. This is demonstrated by the decreasing reaction rate for MHET hydrolysis over time (Supplementary Figure 9a). The effect of product inhibition is likely less pronounced in the natural environment where formed TPA is metabolized by the bacterium. As MHETase was previously not biochemically characterized in detail, temperature and pH profiles were also recorded (Supplementary Figure 9b-c). With high activities from pH 6.0 to 9.5, the enzyme is applicable over a broad pH range. The activity increases with rising temperature up to 44 °C after which the enzyme is rapidly inactivated.

While residues of the catalytic triad were verified by activity assays with the respective alanine mutants, an unaltered high turnover of H488A rules out the presence of a catalytic tetrad in MHETase (Figs. 2b, 4a, b)[24]. The importance of F495 for substrate binding is underlined by significant decreases in turnover rates of the natural substrate MHET and the chromogenic substrate MpNPT by the respective alanine mutant (Fig. 4a, b). The activity at high substrate concentrations is increased with the W397A variant at the expense of a lower substrate affinity (Fig. 4a, b).

While this mutation could be a disadvantage for the bacterium in the natural environment where low substrate concentrations are expected, it is advantageous for biotechnological applications running at substrate concentrations about $10^5$-fold higher than $K_m$—which is in the 100 mM range.

The structural reasons for a high MHETase activity towards MHET ($k_{cat}$ 11.1 ± 1.4 s$^{-1}$) and a very low activity towards BHET ($k_{cat}$ 0.0011 ± 0.0002 s$^{-1}$) have not been explained before (Fig. 4b, c)[10]. In the light of our structural data, we anticipated that modifications in the distal part of the binding pocket that mediates electrostatic interactions with the MHET carboxylate may confer activity towards BHET. Strikingly, the S416A and S419G mutants retain MHETase activity and permit the conversion of BHET to TPA which may be explained by the increased flexibility of R411 in these mutants allowing BHET binding (Fig. 4c). Also, providing more space in the inner active site and introduction of potential hydrogen bonding partners conferred by the variants F424Q and F424N significantly increases the turnover of BHET by MHETase (Figs. 2b, 4c). The removal of positive charge in the variants R411A and R411Q also allows a significantly higher turnover of BHET. When these

mutations are further combined with mutations at S416 and S419, the turnover of BHET can be increased 120-fold compared to wild type (Fig. 4c).

Lastly, wild-type MHETase and variants S416A F424N and R411A S419G F424N, which have BHETase activity, were also examined for activity towards coumaric acid methyl ester, caffeic acid methyl ester, chlorogenic acid and p-hydroxy benzoic acid methyl ester, substrates for feruloyl and chlorogenate esterases. No activity above background (no enzyme) could be detected.

## Discussion

The microbial degradation and metabolization of PET—and its degradation intermediate MHET—as a carbon and energy source has only come up recently in the environment. To understand the evolutionary origin of MHETase is therefore highly relevant for enzymatic plastic degradation in general[10,18]. Phylogenetic analysis groups MHETase in the tannase family of the α/β-hydrolase fold enzymes (Supplementary Figure 4). Its closest relatives act on the larger substrate ferulate whereas the more similar gallate is the substrate of more distantly related tannases. Whether MHETase derived from a feruloyl esterase or tannase ancestor cannot be answered yet. Typical tannase substrates like hydroxy cinnamoates and hydroxy benzoates are neither converted by MHETase (as shown already[10]) nor by the MHETase variants with BHETase activity which we engineered and presented here.

Our structural results on MHETase identified the lid domain as the major difference to the closely related tannase and feruloyl esterases (Fig. 1b–e). Interestingly, it was already pointed out that major switches in enzyme function might occur during natural evolution through loop insertion, deletion or recombination[26]. To this end, our results indicate that MHETase might originate indeed from a loop modification in the lid domain leading to the reported activity towards MHET hydrolysis although no homologous tannase or feruloyl esterase sequence could be identified in the *I. sakaiensis* genome. Notably, these loops confer a crucial specificity for the *para*-carboxy group of the substrate (Fig. 3). This specialization for the natural substrate MHET also explains the very low activity of MHETase towards the intermediate BHET in the wild-type enzyme. In the natural environment, missing activity towards BHET is not critical as the upstream enzyme, PETase, already hydrolyzes BHET to MHET[10].

The structure of MHETase represents a key step in understanding the process of microbial PET degradation in *I. sakaiensis*. Our structural and mutational analyses shed light on the substrate recognition using an induced-fit mechanism and enabled first structure-guided alterations of substrate specificity of MHETase. We were thus successful in generating an MHETase variant, which hydrolyzes the PETase products MHET and BHET down to the very building blocks, which are required for a sustainable re-synthesis of the polyethylene terephthalate polymer. Contrasting consensus α/β-hydrolases, the bipartite architecture of MHETase parts catalysis from substrate recognition—a scenario where we envision the lid domain as a tunable platform to enhance catalytic properties (e.g. alleviating substrate release) or alter substrate specificity (as shown initially for the S416A, R411Q or F424N mutants reported here).

With the structures of MHETase available, our detailed insights into its mechanism and in particular the generation of a BHETase with altered substrate specificity, it will now be possible to rationally create even more efficient MHETase variants cleaving other partial degradation products from related polymers. Replacing TPA in PET by thiophen-, furan- or pyridine-dicarboxylic acid has long been described[27]. Exchanging the carboxylic esters by sulfonic esters in polymers is also possible[28]. Polyesters of 2,5-furandicarboxylate with ethylene glycol (PEF) or other alcohols are suitable to replace PET in bottles[29]. This new plastic PEF can be degraded by PETase[19]. The product hydroxyethyl-2,5-furandicarboxylate is similar enough to MHET to envision structure-guided mutagenesis of MHETase to evolve a "MHEFase" for the full cycle from renewable carbohydrates to PEF and back to polymer building blocks by green chemistry.

The potential use of MHETase in recycling of these alternative polyesters underlines the need to understand and customize binding to different substrates. We thus anticipate that our extensive structural characterization and initial rational modulation of MHETase substrate specificity provides an excellent starting point for the development of tailor-made, enzymatic PET degradation systems based on a MHETase scaffold and in combination with PETase.

## Methods

**Reagents**. PET was obtained from commercial bottles. All other chemicals were purchased at the highest purity from Sigma-Aldrich, Carl Roth, Alfa Aesar or Acros if not stated otherwise.

**Synthesis of ligands and substrates**. Identity and purity of all synthesized compounds was verified by NMR. $^1$H spectra were measured in DMSO-$d_6$ on a Bruker Avance II 300 equipped with a 5 mm PABBO BB-1H/D Z-GRD Z104275/0398 probehead at 25–28 °C (Fig. S7a-f). For the calibration of the measurements tetramethylsilane was used.

*Bishydroxyethyl terephthalate (BHET)*: BHET was synthesized from a PET bottle by alcoholysis with ethylene glycol. Twenty grams PET and 0.2 g anhydrous sodium acetate were refluxed in 120 mL ethylene glycol for 8 h and afterwards cooled overnight. 120 mL $H_2O$ was added and filtration was performed at 4 °C. The product was washed with 20 mL cold $H_2O$ and extracted several times with hot $H_2O$. BHET appeared as white needles (18 g (68%), Mp 210–212 °C).

*Bishydroxyethyl terephthalic acid amide (BHETA)*: BHETA was synthesized from PET by aminolysis with 2-amino ethanol. Twenty grams PET and 0.2 g anhydrous sodium acetate were refluxed in 120 mL ethanolamine for 8 h and afterwards cooled overnight. 120 mL $H_2O$ was added and filtration was performed at 4 °C. The product was washed with 20 mL cold $H_2O$ and recrystallized twice with 100 mL hot $H_2O$. BHETA appeared as lightly rose needles (20 g (76%), Mp 240–243 °C).

*Dimethyl terephthalate (DMT)*: DMT was synthesized by esterification of terephthaloyl chloride with methanol. 25 mmol terephthaloyl chloride was reacted with 30 mL methanol at RT and then refluxed for 3 h. After distilling the methanol and drying at 60 °C, 4.08 g was obtained (Mp 144–148 °C). Washing with 0.5 M KOH and water did not change the melting point.

*Monohydroxyethyl terephthalate (MHET)*: MHET was synthesized from BHET by partial hydrolysis with KOH. 8.7 mmol BHET was reacted with 8.4 mmol KOH in 18 mL MgSO$_4$-dried ethylene glycol at 110–130 °C for 2.5 h. Thirty milliliters $H_2O$ was added and the mixture was extracted three times with 5 mL CHCl$_3$. The aqueous phase was adjusted to pH 3 with 25% HCl and filtered at 4 °C. After two extraction steps with 30 mL hot $H_2O$ and filtration at 4 °C, the precipitate was dried at 60 °C (0.56 g (30%), Mp 185–190 °C).

*Monohydroxyethyl terephthalic acid amide (MHETA)*: MHETA was synthesized by partial amidation of terephthaloyl chloride with ethanolamine. 150 mmol NaOH and 50 mmol ethanolamine in 50 mL $H_2O$ were added dropwise within 1 h to 50 mmol terephthaloyl chloride in 50 mL $H_2O$ at 0 °C. The reaction was performed for another 2 h at 0 °C and 2 h under reflux, followed by hot filtration. The pH was adjusted to 3 with 25% HCl. The obtained suspension was filtered cold and the filtrate was washed with 20 mL cold water. The product was recrystallized from 100 mL hot $H_2O$ to yield shiny crystals (2.4 g (23%), Mp 209–212 °C).

*Mono-4-nitrophenyl terephthalate (MpNPT)*: MpNPT was synthesized by esterification of terephthaloyl chloride with 4-nitrophenolate. 50 mmol terephthaloyl chloride and 50 mmol sodium 4-nitrophenolate were suspended in 50 mL diethylether and reacted for 2 h at 0 °C, then at RT overnight. 2.5 g Na$_2$CO$_3$ and 4.5 g NaHCO$_3$ in 50 mL $H_2O$ were added and reacted at RT for 10 h. The pH was adjusted to 8.5 with NaOH. The insoluble fraction was further extracted with a total of 2.5 g Na$_2$CO$_3$ and 2.5 g NaHCO$_3$ in 100 mL $H_2O$ and then washed until a neutral pH. MpNPT was precipitated with HCl at pH 3 and washed twice with 50 mL 0.1 M HCl and then until a neutral pH. MpNPT was separated from contaminating bis-4-nitrophenyl terephthalate by extraction with 100 mM NaPi pH 7.4 and acid precipitation. The very faint yellow slurry was dried at 60 °C (Mp 202 °C).

**Purification as well as crystallization and structure solution**. *I. sakaiensis* PETase (amino acid residues 28–290) was ordered from Genscript (Piscataway,

USA) as a codon-optimized synthetic gene containing a C-terminal His$_6$-tag subcloned into pET-21b. A codon-optimized DNA fragment encoding *I. sakaiensis* MHETase (amino acid residues 20–603) cloned in a pUC19 vector was ordered from Genscript and later subcloned into a pColdII expression plasmid with a N-terminal His$_6$-tag (TAKARA BIO, Inc., Otsu, Shiga, Japan) by FastCloning (Supplementary Figure 11)[30].

For protein expression, *E. coli Shuffle* T7 express cells (New England Biolabs, Frankfurt, Germany) were transformed with the plasmids and selected on lysogeny broth (LB) agar plates containing 100 µgmL$^{-1}$ ampicillin. After growth overnight at 30 °C, overnight cultures were inoculated. For overexpression, 1 L baffled Erlenmeyer flasks containing 200 mL LB medium supplemented with 100 µg mL$^{-1}$ ampicillin were used. The cells were grown at 33 °C and 160 rpm shaking velocity to an optical density at 600 nm (OD$_{600}$) of 1 before 1 mM isopropyl β-D-1-thiogalactopyranoside (IPTG) was added. At an OD$_{600}$ of 2.5, the temperature was lowered to 16 °C and the overexpression continued overnight. The cells were harvested by centrifugation at 4 °C, 10,000 × $g$ for 20 min and stored at –20 °C until further use.

Cells were disrupted by sonication in 50 mM TRIS-HCl, pH 7.5, 150 mM NaCl, 10 mM imidazole and 1 mM Dithiothreitol (DTT) (buffer R). Cell debris was cleared by centrifugation. The cell extract was loaded on a gravity flow column with Ni-NTA sepharose, washed with buffer R supplemented with 20 mM imidazole and eluted with buffer R supplemented with 200 mM imidazole (300 mM imidazole in the case of PETase). Protein fractions were purified on a Superdex75 10/300 column (GE Healthcare, Solingen, Germany) with 20 mM TRIS pH 7.5, 150 mM NaCl, concentrated to ~10 mg mL$^{-1}$, flash-frozen in liquid N$_2$ and stored at –80 °C. Protein concentrations were determined spectrophotometrically using ε$_{280}$ = 102 955 M$^{-1}$ cm$^{-1}$ (ε$_{280}$ = 97,455 M$^{-1}$ cm$^{-1}$ for mutant W397A). Purity of the samples was estimated employing the GelAnalyzer software (Version 2010a).

PETase was crystallized at 10 mg mL$^{-1}$ concentration by sitting drop vapor diffusion (1 µL protein plus 1 µL reservoir) at 20 °C. Typical PETase crystals grew at 20 °C with a reservoir solution containing 0.1 M sodium citrate or sodium acetate pH 5.0, 15% (v/v) PEG8000 and 0.5 M lithium sulfate. MHETase crystallized at a concentration of 10 mg mL$^{-1}$ by sitting drop vapor diffusion (1 µL protein plus 1 µL reservoir) at 20 °C with a reservoir containing 0.1 M HEPES, pH 7.5, 30% (v/v) 2,4-MPD and 0.12 M ammonium sulfate (space group P2$_1$2$_1$2$_1$) or 0.1 M MES, pH 6.5, 10% (v/v) PEG8000 and 0.1 M zinc acetate (space group P1). MHETase crystals grown with MPD were cryo-protected in their reservoir solution. The PETase crystal was cryo-protected with 0.1 M sodium acetate, pH 5.0, 10% (v/v) PEG8000, 15% (v/v) PEG400 and 0.5 M lithium sulfate. The P1 MHETase crystal was cryo-protected with 0.1 M TRIS, pH 8.5, 5% (v/v) PEG8000, 20% (v/v) PEG400 and 0.5 M lithium sulfate. For derivatization experiments, crystals were incubated for 24 h in the respective cryo-protectant solution saturated with the ligand and flash-frozen in liquid nitrogen. MHETase diffraction data were collected at 100 K at beamlines 14.1 and 14.2 of the BESSY II storage ring, Berlin, Germany[31]. PETase diffraction data were collected at beamline P13, PETRAIII, Hamburg[32]. All diffraction data were processed with XDS[33,34].

The structure of PETase was solved by molecular replacement employing the structural coordinates of *T. fusca* cutinase TfCut2 (PDB entry 4CG1[11,23]). The structure of MHETase complexed to MHETA was solved by molecular replacement employing the MR pipeline MoRDa[35], which placed six copies of a homology model based on PDB entry 6G21 (an unpublished *A. oryzae* feruloyl esterase, AoFaeB-2; 26% identity to MHETase with a query coverage of 87%). Structures were complemented during several rounds of refinement with PHENIX. REFINE or Refmac in the case of PETase[35,36] intermitted by manual model building with COOT which was extensive for MHETase[37]. The structure of MHETase complexed to BA was solved by molecular replacement employing the entire asymmetric unit of the MHETA co-structure as a search model and completed, respectively. The MHETA and BA ligands were placed into the respective densities unambiguously at the final stages of refinement. The structure of the P1 ligand-free MHETase was solved by molecular replacement employing structure coordinates of a finalized MHETase-MHETA co-structure omitting the ligand. The ten copies of MHETase in the asymmetric unit were built manually or by PHENIX.AUTOBUILD and refined with PHENIX.REFINE[36].

**Generation of MHETase mutants by site-directed mutagenesis**. For the creation of single-site mutants, either the Q5 site-directed mutagenesis kit (New England Biolabs) or the QuikChange® method were used. In the first case, the kit was used according to the manufacturer's instructions. In case of the QuikChange® method, a standard PCR mixture (25 µL) consisting of 2.5 µL 10× Pfu buffer (Roboklon GmbH, Berlin, Germany), 0.5 µL mixed deoxynucleoside triphosphates (0.25 mM each), plasmid DNA (~40 ng), 1.25 µL forward and reverse primers (10 µM; Supplementary Table 4), 1 µL PfuPlus! polymerase (Roboklon GmbH) and 18.1 µL ultrapure water was used. For denaturation, the temperature was held at 95 °C for 30 s. Afterwards 20 cycles of 30 s denaturation at 95 °C, annealing for 30 s at 63 °C and elongation for 6 min at 72 °C were conducted. In a last step the final elongation was accomplished at 72 °C for 10 min. After PCR, the template DNA was digested with *Dpn*I (New England BioLabs) for 2 h at 37 °C before the enzyme was inactivated at 80 °C for 10 min. *E. coli* TOP10 cells were transformed with the obtained PCR products and plated on LB agar plates containing 100 µg mL

$^{-1}$ ampicillin and grown overnight at 37 °C. The nucleotide sequence was confirmed by sequencing at Eurofins (Ebersberg, Germany).

**Autohydrolysis of MpNPT**. Hydrolysis was followed spectrophotometrically at 400 nm at 30 °C. At higher pH values, where hydrolysis goes to near completion, exponential decay was fitted. At lower pH values decay constants were calculated from initial rates taking into account the purity and the partial deprotonation of 4-nitrophenol (the expected value at pH 7.5 is 12,420 M$^{-1}$ cm$^{-1}$ calculated with ε$_{405\ nm}$ = 18,000 M$^{-1}$ cm$^{-1}$ and pK$_a$ = 7.15 for 4-nitrophenol)[38]. Absorption difference of 4-nitrophenol at 400 and 405 nm was less than 0.5%. Autohydrolysis was measured from pH 7.5 to 12.5 in 100 mM phosphate or borate buffers and NaOH solutions with a rate constant of $k_2$ = 9.2 M$^{-1}$ s$^{-1}$ for d[MpNPT]/dT = $k_2$ [OH$^-$] [MpNPT], i.e. $k_1$ = 2.9 × 10$^{-6}$ s$^{-1}$ (Supplementary Figure 12). In 100 mM TRIS pH 7.5, also used for enzyme kinetics, $k_1$ = 46 × 10$^{-6}$ s$^{-1}$ for d[MpNPT]/dT = $k_1$ [MpNPT] was determined. TRIS thus increases the hydrolysis rate ~10-fold compared to the expected reaction rate with hydroxide at pH 7.5.

**Activity tests**. Activity was measured for the substrates MHET and BHET by HPLC and for MpNPT by spectrophotometry. For the HPLC measurement, the reactions (100 µL) were stopped by addition of an equal volume of 160 mM NaPi pH 2.5 with 20% (v/v) dimethyl sulfoxide (DMSO) and heating to 80 °C for 10 min. TPA, MHET and BHET were separated on a Kinetex 5 µm EVO C18 100 Å, 150 × 4.6 mm (Phenomenex, Aschaffenburg, Germany) with a gradient of acetonitrile and 0.1% (v/v) formic acid in water at 30 °C after injection of 10 µL sample. Acetonitrile was increased from 5 to 44% until minute 12 and then to 70% at minute 15 where the ratio remained constant for 3 min. TPA, MHET and BHET were detected at 240 nm and quantification was realized by the use of calibration curves.

To quantify the turnover rates of MHET and BHET, reactions (100 µL) with 1 mM of the substrate in 40 mM NaPi pH 7.5 with 80 mM NaCl and 20% (v/v) DMSO were incubated at 30 °C before they were stopped and analyzed as described above. MHETase was used at 6 nM for 30 min in case of MHET. For BHET, a prescreening with 6 nM or 12 nM MHETase variants was used to identify potentially active enzymes which were afterwards analyzed at higher concentrations. Around 100 nM of the variants were used for 19.25 h and 3 nM wild-type MHETase was added to these reactions for the full conversion of the formed MHET to TPA which was quantified via HPLC. The experiment was repeated three times.

MpNPT stock solutions were prepared in DMSO at concentrations of 10, 1 and 0.1 mM. Substrate concentrations were 0.1–1200 µM in 100 mM TRIS pH 7.5 or 100 mM NaPi pH 7.5. Kinetic parameters are the same in both buffers. The reaction in the 400 µL cuvette was started at 25 °C by addition of the enzyme to a final concentration of ca. 1 nM (active variants) up to 500 nM (inactive variants). Absorption change was followed on a Cary50 spectrophotometer (Varian) at 400 nm. Absorption changes were corrected for nonenzymatic hydrolysis (see previous paragraph). Initial rates (on average ten different combinations of enzyme and substrate concentrations for each variant) were used for fitting $K_m$ and $v_{max}$ according to the Michaelis Menten kinetics. $K_I$ for competitive inhibitors were measured at 30 °C and are calculated with Eq. (1)

$$v = v_{max} * [S] / \left( [S] + K_M \left( 1 + \frac{[I]}{K_I} \right) \right) \qquad (1)$$

using 4−8 different substrate (i.e. MpNPT) concentrations. Wild type and variants S225A, 488A, W397A, F495A, F415A, S416A and R411Q were also measured at 30 °C. These kinetic parameters were normalized on wild-type activities at 25 °C.

For measurement of feruloyl and chlorogenate esterase activity four substrates were tested: coumaric acid methyl ester (Coum-ME), caffeic acid methyl ester (Caff-ME), chlorogenic acid (Chlorogen) and *p*-hydroxy-hydroxy benzoic acid methyl ester (pHB-ME). UV−Vis spectra using 10 µM of the ester and the free acid were measured and Δε calculated (Supplementary Fig. 10). Hydrolysis was measured as for MpNPT, but with 10–35 nM enzyme, 100 µM substrate and at 335 nm (Coum-ME, Δε = −6100 M$^{-1}$ cm$^{-1}$), 350 nm (Caff-ME, Δε = −5700 M$^{-1}$ cm$^{-1}$), 350 nm (Chlorogen, Δε = −7400 M$^{-1}$ cm$^{-1}$) and 280 nm (pHB-ME, Δε = −3900 M$^{-1}$ cm$^{-1}$).

**Differential scanning fluorimetry**. For the analysis of ligand binding to MHETase, DSF was used. Experiments were conducted with a Prometheus NT.48 (NanoTemper, Munich, Germany). The device has a fixed excitation wavelength of 285 nm and emission wavelengths of 330 and 350 nm. MHETase (wt) was always used at 100 µgmL$^{-1}$ in the final solution. Final buffer concentrations were 100 mM TRIS pH 7.5, 150 mM NaCl with or without 20% DMSO. High sensitivity capillaries as provided by NanoTemper were used. The temperature range 20 to ≥80 °C was scanned at 0.5 K per min. Ligands were prepared in 21.7 mM stock solutions and diluted to 10 mM final concentration. The saturated solution (with 0 or 42.5% DMSO) was used as stock, if the compounds did not fully dissolve. For compounds hindering reliable measurements by absorption (4-nitrophenol, 4-nitrothiophenol, 2-hydroxybenzoic acid) or fluorescence (BHET) 1 mM final concentrations were also tested. Tm values are reported as provided by the Prometheus software

(maximum of the slope for the $I_{330\,nm}/I_{350\,nm}$ ratio). The experiment was performed as a single measurement.

**Sequence alignment and phylogenetic tree.** Protein homology searches for MHETase-like proteins were performed with the NCBI basic local alignment search tool (BLAST) in the ESTHER database (http://bioweb.ensam.inra.fr/ESTHER/general?what=blast) using the Block_X.pep database[22,39]. Multiple sequence alignment was performed by Muscle using MEGA7[40]. The evolutionary history was inferred by using the Maximum Likelihood method based on the JTT matrix-based model[41]. The tree with the highest log likelihood (−19,562.87) is shown. Initial tree(s) for the heuristic search were obtained automatically by applying Neighbor-Join and BioNJ algorithms to a matrix of pairwise distances estimated using a JTT model, and then selecting the topology with superior log likelihood value. The analysis involved 32 amino acid sequences. All positions containing gaps and missing data were eliminated. There was a total of 376 positions in the final dataset. Evolutionary analyses were conducted in MEGA7[40].

## Data availability

Structure coordinates and diffraction data were deposited with the Protein Data Bank (http://www.pdb.org) under accession codes 6QG9 (MHETase), 6QGA (MHETase MHETA), 6QGB (MHETase BA), 6QGC (PETase). The source data underlying Figs. 3, 4a-c, and Supplementary Figures 2a-b, 4, 8, 9a-c and 10 is provided as a Source Data file. Other data are available from the corresponding authors upon reasonable request.

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

## Acknowledgements

We thank Leonie Graf for excellent experimental help. We acknowledge access to beamlines BL14.1/2/3 of the BESSY II storage ring (Berlin, Germany) via the Joint Berlin MX-Laboratory sponsored by the Helmholtz Zentrum Berlin für Materialien und Energie, the Freie Universität Berlin, the Humboldt-Universität zu Berlin, the Max-Delbrück Centrum, and the Leibniz-Institut für Molekulare Pharmakologie. The PETase data were collected at beamline P13 operated by EMBL Hamburg at the PETRA III storage ring (DESY, Hamburg, Germany). We would like to thank Isabel Bento, Gleb Bourenkov, and Thomas Schneider for their assistance in using the beamline.

## Author contributions

U.T.B and G.W. initiated the study and directed the project. M.C.W, L.B., L.R., E.A.P.M. and H.M. cloned, expressed and purified PETase, MHETase and mutants, conducted binding studies and activity assays under the supervision of D.B. and G.J.P. E.A.P.M. and G.J.P. synthesized MHET analogs. E.A.P.M., G.J.P., M.S.W. and G.W. conducted crystallographic and computational analyses. G.J.P, M.S.W., U.T.B. and G.W. prepared the manuscript, which was revised and approved by all authors. This work was supported by a startup funding grant from the University of Greifswald to G.W.

**Additional information**

**Competing interests:** The authors declare no competing interests.

