## [Peer Review File · Nature Communications]

Reviewers' comments:

Reviewer #1 (Remarks to the Author):

The manuscript describes structure determination and mutagenesis studies to reveal the structural basis of substrate recognition and reaction mechanism of mono-(2-hydroxyethyl) terephthalate (MHET) hydrolase (MHETase) from *Ideonella sakaiensis*, which plays the "second" key role in the recently found bacterial PET degradation pathway (Yoshida et al., 2016; this work is the hallmark). This is the 3rd structural report of the Tannase family proteins in the ESTHER database (see below). However, the sequence similarities with the structure-known fungal feruloyl esterases in the Tannase family are very low (~26% amino acid sequence identity), highlighting its scientific impact in the structural biology field. In addition, the mutagenesis study clarified the important residues in the structure recognition, and the authors successfully changed its substrate specificity toward BHETase. Because the authors did not correctly describe the situation of the protein families, the reviewer asks them to thoroughly rewrite the introduction and structural comparison sections.

Major points:

1. In this paper, the protein families and similarities of the proteins (enzymes) are not clearly recognized. The authors indicate a phylogenetic tree in Fig. S4 but it contains so many uncharacterized proteins. Phylogenetic analysis and classification of the alpha/beta-hydrolase fold proteins is not an easy task for non-specialists because of frequent indels, circular permutations, domain exchanges, and low positional conservation of the second and third catalytic residues of the triad in the sequence. The reviewer strongly suggests the authors to consult the ESTHER classification database (<http://bioweb.enscm.inra.fr/ESTHER/general?what=index>), which is dedicated to wide array of the alpha/beta-hydrolase fold proteins.
2. MHETase belongs to the "Tannase" family in the ESTHER database. In the family, structures of two fungal esterases have been available on PDB (3WMT for FaeB from *A. oryzae* and 6G21 for an unknown esterase from *A. oryzae*; 6FAT is still on hold). Even though the second entry (6G21, referred AoFaeB-2 in the text) is an "unpublished" result, it is used as the structural comparison pair in Fig. 1 and the main text whereas the paper of the entry 3WMT (ref #22; Suzuki et al., 2014), which is the first and sole report for Tannase family structure, is not appropriately cited. Detailed structural and sequence comparisons (Fig. 1 and Fig. S1) with the "published" Tannase family protein structure is required.
3. The authors seem to confuse the two distinct tannase-containing families. The tannase from *Lactobacillus* species belongs to the "Tannase_bact" family in Block H, which is clearly different from the mainly fungal "Tannase" family in Block X (see the ESTHER database). Situation of these two families must be clearly described in the Introduction section. However, structural comparison with the ethyl gallate complex of LptE (4JOK) is worth doing (Fig. 2 and Fig. S3).
4. In the ref #22, it has been shown that the Tannase family proteins have a unique CS-D-HC motif. The description about the disulfide bond near the catalytic triad (P4L11-12) is insufficient. At least, structural comparison around the disulfide bond and the catalytic triad must be shown.
5. The core fold of most esterases is commonly called "alpha/beta-hydrolase fold" (Nardini & Dijkstra, *Curr. Opin. Struct. Biol.* 9, 732-373, 1999), not "a hydrolase fold" (P4L8), "a feruloyl esterase fold" (Fig.1 legend) or "the core hydrolase fold" (P5L9).
6. P5. The two descriptions in L11 "The MHETase-MHETA complex structure is nearly identical to the apo structure.." and in L21 "... reveals an induced-fit mechanism upon MHETA binding" appear to be mismatched. The apparent mismatch must have been derived from confusion about the main-chain or side-chain conformational deviations.
7. Figure 1C and Fig. S3a. Superimposition of more than two ribbon models is too complicated. These panels are better to be shown as side-by-side comparison.
8. Abstract (P2L2). Although the reviewer is not a native English speaker, the Abstract starting with the word "Its" looks unusual in scientific papers.
9. P3L9-12. Add a new figure showing the degradation scheme of PET by *I. sakaiensis*.
10. P4L2. Around 2.0 Å is not high-resolution structures.

Minor points:

1. Title (P1). *I. sakaiensis* -> *Ideonella sakaiensis*
2. Abstract (P2L12). Substrate requirements -> substrate specificity ?
3. The references #16 and #21 are identical.
4. P5L15. ... a lesser extent G132 (?)
5. P11L9. 100000-fold is better to be described as 10⁽⁵⁾-fold (using superscript).

Reviewer #2 (Remarks to the Author):

The authors present the structure of the MHETase enzyme, which catalyses the second step of PET degradation, and is clearly therefore a topic of interest. With the highest sequence ID homolog available in the PDB of 26%, the structure is also sufficiently novel to suggest diverged evolution of function, and to merit the initiation of detailed examination described in this manuscript.

Overall, the paper is well presented and written, although some sentences could be refined. There are also some Nature Comms protocols that should be adhered to in a revised manuscript - including the presentation of stereo figures for representative density and the deposition of data in order to present accessions codes. Some comments on the manuscript follow

1. Abstract – ‘The extreme durability of PET debris has rendered it..’
2. Abstract Line 9 ‘was not reported’ – rather than ‘could not be elucidated’
3. Abstract Line 10 rephrase ‘domain exerts catalysis’ better to say that ‘MHETase possesses a classic a/b fold’ etc.
4. Abstract line 11 – the ‘lid domain’ does not itself resemble feruloyl esterases – do you mean ‘resembling that of feruloyl esterases’?
5. The last line should be rephrased to indicate that these experiments are included in the paper i.e. ‘The structure has been used to inform...’, ‘which will serve as a basis for engineering MHETase for enzymatic plastic degradation’ or similar.
6. Intro Line 21 ‘...moiety, rendering it..’
7. Page 3 line 1 ‘they are also’
8. Page 3 line 2 – please define/expand ‘PET layers’
9. Page 3 line 13 should read ‘mode of catalysis’ or ‘residues responsible for catalysis’ etc. and then ‘even permitted the enhancement of..’
10. Page 3 line 16 – what are ‘clear’ homologs – state limits of sequence ID
11. Page 3 line 21 the enzyme does not ‘harbor’ a scaffold – please rephrase
12. Line 22 – ‘..and expand substrate..’
13. Page 4 line 6 – state which structure (PDB code) and sequence ID
14. Line 7 - the lid domain is not ‘interspersed’ please rephrase
15. Line 9-10 – the presence of Ca is confirmed by X-ray FS, but not the ‘site’; please rephrase
16. Line 12 ‘backbone amide nitrogens of G132..’
17. Page 5 line 13 ‘substrate specificity’
18. Line 17 remove ‘specific’
19. Page 6 line 15 – ‘very tightly’ please give comparative values of K_m presumably
20. Page 7 line 6 rephrase ‘addressed’
21. Rephrase line 8 ‘Contrasting MHETase..’ – do you mean: ‘By contrast, in MHETase, ‘ (?)
22. Line 14-15 ‘or pNP respectively’
23. Page 9 line 1 – A comparison of ...(what?) of potential ligands..’
24. Line 14 and 18 – check notation of K_m – two different uses here
25. Line 21 ‘also the likely cause’
26. Page 12 Line 12 – the evolution question posed here does not seem to belong to the discussion as it stands
27. Line 25 – this suggestion seems to demand data – is the activity of MHETase with this substrate known? Otherwise ‘alternative polyesters’ is probably a better general statement to finish with.

28. Page 16 – Line 19 – state what this structure is (6G21)
29. References –there are some inconsistencies in page numbering etc. e.g. 26, 37
30. The structures are not deposited- the validation reports submitted clearly state that those reports are not to be submitted to journals - PDB codes must be included in a revised submission.
31. Nature Comms asks that 'To facilitate assessment of the quality of the structural data, a stereo image of a portion of the electron density map' [should be provided] – it is suggested that this is done for the density figure in the SI.
32. Please check significant figures for the Crystallography Data Table – resolution, R-factors (3 dec places or 22.1% etc.), B factors (integer values)
33. Please provide separate B-factors for solvent and ligands

Reviewer #3 (Remarks to the Author):

Comments

This manuscript describes very extensively the structural investigation of *I. sakaiensis* MHETase which degrades in combination with *I. sakaiensis* PETase the synthetic polymer polyethylene terephthalate. Based on structural data, MHETase mutants were generated showing activity on BHET which is totally missing in wt MHETase. However, essential data are missing showing the effect of generated mutants in combination with PETase. How much BHET is formed by PETase and how significant is the effect of BHET activity for total PET degradation? Additionally, the section Materials and Methods has to be completely revised since important information is missing and several mistakes are in the text (see some comments below).

Major comments

Introduction, line 10: it is stated in line 6, that so far no enzyme has been found that fully penetrates and degrades a thick layer of highly crystalline PET in a cost-effective manner. What is the performance of the two enzymes, PETase and MHETase, compared to the already known and published enzymes? How much PET is degraded and in which time? What`s the crystallinity of the PET used in the studies?

Page 3, line 16: assignment to the tannase family and sequence homology to the feruloyl esterases from *A.oryzae* should be confirmed by testing MHETase with selected substrates of this enzyme family.

Page 3, line 2: what are the homologies for the hydrolase domains?

Page 4, line 6: how high is the homology between MHETase and *A.oryzae* feruloyl esterase?

Page 4, line 9: what is the function of the calcium ion?

Figure 1: is quite overloaded and repetitive, select and display most important information

Page 5, line 22: the term "apo-MHETase" is quite confusing since apo-enzymes are defined as enzymes lacking a cofactor. In this manuscript, the term is used for description of the free enzyme in contrast to the enzyme-substrate-complex?

Page 9, line 24: what do you mean with "comparably low activity of MHETase"? With which enzymes and activities do you compare the MHETase activities?

Page 11, line 11: it is not explained why the missing activity on BHET is of importance; how much BHET is released in the course of PET degradation by the PETase? Is BHET the main product?

Page 12, line 1: structural investigation of MHETase is a "key step in understanding the full process of microbial PET degradation by *I. sakaiensis*" but cannot be generalized.

Page 12, line 6: it is stated that MHETase variants can degrade several PETase products down to monomers; what do you mean with "several PETase products" except of MHET and BHET?

Page 12, line 17: also for the evolution of MHETase from tannase it would be essential to verify MHETase activity towards tannase substrates.

Page 12, line 23: which polymers do you have in mind that contains heteroaromatics or sulfonic acid groups?

Page 13, line 10: what is the degree of crystallinity? Since it is described in the introduction that *I. sakaiensis* can degrade high crystalline PET this information is important.

Page 13 and 14: chemical and structural analysis of synthesized molecules as well as determination of purities are fully missing.

Page 16: here, the crystallization and analysis of PETase is described, but in the text it is described that this is another publication?

Page 18, line 8: what was the reaction mixture? Provide details regarding enzyme concentration, reaction volume, substrate concentration, reaction conditions etc.

Minor comments

Page 5, line 6: give the full name for *L. plantarum* since it is mentioned for the first time in the manuscript

Page 7, line 10: provide the full name for HEMT and pNP

Response to Reviewer Comments

In the revised version of our manuscript, we have addressed all points (changes are highlighted with **yellow color**.) raised by the reviewers as detailed below. We have also characterized a new series of MHETase mutants which now exhibit a significant BHETase activity (as shown in Fig. 4c) and which corroborate our prior data. Lastly, we have implemented a plethora of textual improvements and rearranged the passage addressing the evolutionary aspects of MHETase in the results part and as well in the supplement. In our rebuttal given below, reviewer comments are repeated in **bold**, responses are in regular font, original and changed text passages are in *italics*. We thank all reviewers for the very fast review process and the highly constructive and valuable assessment of our manuscript.

Reviewer #1

The manuscript describes structure determination and mutagenesis studies to reveal the structural basis of substrate recognition and reaction mechanism of mono-(2-hydroxyethyl) terephthalate (MHET) hydrolase (MHETase) from Ideonella sakaiensis, which plays the “second” key role in the recently found bacterial PET degradation pathway (Yoshida et al., 2016; this work is the hallmark). This is the 3rd structural report of the Tannase family proteins in the ESTHER database (see below). However, the sequence similarities with the structure-known fungal feruloyl esterases in the Tannase family are very low (~26% amino acid sequence identity), highlighting its scientific impact in the structural biology field. In addition, the mutagenesis study clarified the important residues in the structure recognition, and the authors successfully changed its substrate specificity toward BHETase. Because the authors did not correctly describe the situation of the protein families, the reviewer asks them to thoroughly rewrite the introduction and structural comparison sections.

We thank the reviewer for highlighting the scientific impact of our study within the structural biology field and appreciating our engineering results towards the structure-based generation of a BHETase. We are also grateful for clarifying uncertainties with the phylogeny of tannases / feruloyl esterases. We have now resolved all points regarding the taxonomic classification of MHETase and feruloyl esterases / tannases as stated in the text below.

Major points:

1. In this paper, the protein families and similarities of the proteins (enzymes) are not clearly recognized. The authors indicate a phylogenetic tree in Fig. S4 but it contains so many uncharacterized proteins. Phylogenetic analysis and classification of the alpha/beta-hydrolase fold proteins is not an easy task for non-specialists because of frequent indels, circular permutations, domain exchanges, and low positional conservation of the second and third catalytic residues of the triad in the sequence. The reviewer strongly suggests the authors to consult the ESTHER classification database (<http://bioweb.ensam.inra.fr/ESTHER/general?what=index>), which is dedicated to wide array of the alpha/beta-hydrolase fold proteins.

The authors agree that the ESTHER database is a better alternative to compare and classify alpha/beta-hydrolase-fold proteins. Therefore, the protein homology search and phylogenetic analysis were repeated using this database, in particular the Block X database.

2. MHETase belongs to the “Tannase” family in the ESTHER database. In the family, structures of two fungal esterases have been available on PDB (3WMT for FaeB from *A. oryzae* and 6G21 for an unknown esterase from *A. oryzae*; 6FAT is still on hold). Even though the second entry (6G21, referred AoFaeB-2 in the text) is an “unpublished” result, it is used as the structural comparison pair in Fig. 1 and the main text whereas the paper of the entry 3WMT (ref #22; Suzuki et al., 2014), which is the first and sole report for Tannase family structure, is not appropriately cited. Detailed structural and sequence comparisons (Fig. 1 and Fig. S1) with the “published” Tannase family protein structure is required.

We agree that it is better to compare MHETase with a published structure like 3WMT. We have now implemented the PDB entry 3WMT in Figures 1 and S1 instead of 6G21. Fig. 1 was improved by a side-by-side comparison of MHETase and FaeB and the respective active sites including the disulfide bond near the catalytic triad (Fig. 1b-e). A description of both overall structures and the respective active sites are now implemented in the main text.

3. The authors seem to confuse the two distinct tannase-containing families. The tannase from *Lactobacillus* species belongs to the “Tannase_bact” family in Block H, which is clearly different from the mainly fungal “Tannase” family in Block X (see the ESTHER database). Situation of these two families must be clearly described in the Introduction section. However, structural comparison with the ethyl gallate complex of LptE (4J0K) is worth doing (Fig. 2 and Fig. S3).

We agree and have now added a paragraph explaining the two distinct families in the introduction section. This text passage reads: *MHETase was initially assigned to the tannase enzyme family, which belongs to Block X of the α/β -hydrolase fold enzymes classified in the ESTHER database^{10,22}. This family includes fungal and bacterial tannases and feruloyl esterases. Other significantly different bacterial tannases can be found in a distinct Block H (Tannases_bact) in this database.'*

4. In the ref #22, it has been shown that the Tannase family proteins have a unique CS-D-HC motif. The description about the disulfide bond near the catalytic triad (P4L11-12) is insufficient. At least, structural comparison around the disulfide bond and the catalytic triad must be shown.

A detailed comparison is now shown in Fig. 1b-e and described in the main text. We have now extended the part on phylogenetic relations by a structural comparison and also a comparison of the active sites. This text passage reads: *'The structures of the closest MHETase relatives are 3WMT and 6G21, two feruloyl esterases of Aspergillus oryzae. With them, MHETase shares not only the catalytic triad S225-H528-D492 (3WMT: S203-H457-D417, 6G21: S169-H421-D381) but also G132 (G125, G91) as part of the anion hole and C224-C529 (C202-C528, C168-C422), whose disulfide bond holds the catalytic residues Ser and His together²⁷. All these residues are in the catalytic domain, the disulfide bond is typical (> 80 % conservation) for the tannase family in Block X of α/β -hydrolases according to the ESTHER database (Fig. 1b-e)^{22,27}. The lid domain of the feruloyl esterases displays the same α -helical fold, but the amino acid sequences cannot be aligned without structural information. A binding pocket – like that for MHET in MHETase - also exists in feruloyl esterases, but not a single one of the lining residues is conserved. However, a comparison of the MHETase active site to that of FaeB reveals several residues around the catalytic triad that may contribute to substrate positioning in a similar fashion (e.g. L235, F354 and L245, F415 in FaeB and MHETase, respectively) (Fig. 1c, e). Most likely, it is the alteration in the substrates, especially the carboxylic acid group of MHET vs. the phenolic (methyl ether) groups and the elongating double bond of ferulates that has evoked this difference. The substrates of tannases, e.g. gallates, are more similar at least with respect to their size to MHET. The only tannase structure available in complex with ethyl gallate is from Lactobacillus plantarum (4J0K), which belongs to the bacterial tannase family in Block H of α/β -hydrolases according to the ESTHER database^{22,26}. The tannase catalytic domains are sufficiently conserved, such that the catalytic triad superimposes well. However, the sequences differ strongly, though, the disulfide bond is missing and the lid domain has a markedly different fold.'*

Also, a short text passage regarding the substrate specificity was added to the discussion: *'The microbial degradation and metabolization of PET - and its degradation intermediate*

MHET – as a carbon and energy source has only come up recently in the environment. To understand the evolutionary origin of *MHETase* is therefore highly relevant for enzymatic plastic degradation in general^{10,18}. Phylogenetic analysis groups *MHETase* in the tannase family of the α/β -hydrolase fold enzymes (Supplementary Fig. S4). Its closest relatives act on the larger substrate ferulate whereas the more similar gallate is the substrate of more distantly related tannases. Whether *MHETase* derived from a feruloyl esterase or tannase ancestor cannot be answered yet. Typical tannase substrates like hydroxy cinnamates and hydroxy benzoates, are not converted by *MHETase* (as shown already¹⁰) nor by the *MHETase* variants with *BHETase* activity which we engineered.

5. The core fold of most esterases is commonly called “alpha/beta-hydrolase fold” (Nardini & Dijkstra, Curr. Opin. Struct. Biol. 9, 732-373, 1999), not “a hydrolase fold” (P4L8), “a feruloyl esterase fold” (Fig.1 legend) or “the core hydrolase fold” (P5L9).

We have exchanged ‘*hydrolase fold*’ to ‘ α/β -*hydrolase fold*’ throughout the entire document. The figure caption of Fig.1 was amended to ‘*I. sakaiensis MHETase displays a bipartite domain architecture resembling that of feruloyl esterases*’, since the lid domain represents a separate folding unit.

6. P5. The two descriptions in L11 “The *MHETase*-*MHETA* complex structure is nearly identical to the apo structure..” and in L21 “.. reveals an induced-fit mechanism upon *MHETA* binding” appear to be mismatched. The apparent mismatch must have been derived from confusion about the main-chain or side-chain conformational deviations.

We agree and have now amended ‘*The MHETase-MHETA complex structure is nearly identical to the apo structure (RMSD 0.54 Å) and sheds light on the positioning of MHET for catalysis*’ to ‘*The main chain conformation in the MHETase-MHETA complex structure is nearly identical to that of MHETase without substrate (RMSD 0.54 Å) and sheds light on the positioning of MHET for catalysis.*’

7. Figure 1C and Fig. S3a. Superimposition of more than two ribbon models is too complicated. These panels are better to be shown as side-by-side comparison.

We agree - all comparisons of structures are now shown side-by side (Fig. 1b,d; Fig. S3a-c)

8. Abstract (P2L2). Although the reviewer is not a native English speaker, the Abstract starting with the word “Its” looks unusual in scientific papers.

We fully agree with the reviewer and have amended ‘*Its extreme durability has rendered polyethylene terephthalate (PET) debris a long-term environmental burden*’ to ‘*The extreme*

durability of polyethylene terephthalate (PET) debris has rendered it a long-term environmental burden. – as also proposed by reviewer #2.

9. P3L9-12. Add a new figure showing the degradation scheme of PET by *I. sakaiensis*.

We appreciate this suggestion and have added the scheme to Fig. 1.

10. P4L2. Around 2.0 Å is not high-resolution structures.

We have now deleted the term '*high-resolution*' and added '*resolution*' to the figures in brackets

Minor points:

1. Title (P1). *I. sakaiensis* -> *Ideonella sakaiensis*

We have replaced '*I. sakaiensis*' by '*Ideonella sakaiensis*' in the title.

2. Abstract (P2L12). Substrate requirements -> substrate specificity ?

We agree, but have replaced '*substrate requirements*' by '*the active site*' to avoid a repetition of '*substrate specificity*' in this sentence.

3. The references #16 and #21 are identical.

The reference Austin *et al.* has now only a single entry in the bibliography (#17).

4. P5L15. .. a lesser extent G132 (?)

Corrected!

5. P11L9. 100000-fold is better to be described as 10⁽⁵⁾-fold (using superscript).

Done!

Reviewer #2

The authors present the structure of the MHETase enzyme, which catalyses the second step of PET degradation, and is clearly therefore a topic of interest. With the highest sequence ID homolog available in the PDB of 26%, the structure is also sufficiently novel to suggest diverged evolution of function, and to merit the initiation of detailed examination described in this manuscript. Overall, the paper is well presented and written, although some sentences could be refined. There are also some Nature Comms protocols that should be adhered to in a revised manuscript - including the presentation of stereo figures for representative density and the deposition of data in order to present accessions codes. Some comments on the manuscript follow

We thank the reviewer for considering our paper well-presented and written as well as pointing out the scientific relevance of our structural studies. We are also grateful for highly valuable textual improvements and thoroughly commenting on sentences that require refinement as well as *Nature Communications* protocols we need to adhere to.

1. Abstract – ‘The extreme durability of PET debris has rendered it..’

We have amended ‘*Its extreme durability has rendered polyethylene terephthalate (PET) debris a long-term environmental burden*’ to ‘*The extreme durability of polyethylene terephthalate (PET) debris has rendered it a long-term environmental burden.*’

2. Abstract Line 9 ‘was not reported’ – rather than ‘could not be elucidated’

We agree and have rephrased the sentence to: ‘*To date, the three-dimensional structure of MHETase has remained elusive.*’

3. Abstract Line 10 rephrase ‘domain exerts catalysis’ better to say that ‘MHETase possesses a classic α/β fold’ etc.

4. Abstract line 11 – the ‘lid domain’ dos not itself resembl[e] feruloyl esterases – do you mean ‘resembling that of feruloyl esterases’?

Both comments refer to the same sentence in the abstract and we agree but still like to highlight the near complete separation of catalysis and substrate binding. We have thus changed the sentence ‘*A classic α/β -hydrolase domain exerts catalysis, while substrate specificity is conferred by a lid domain resembling feruloyl esterases.*’ to ‘*MHETase is built up of a classic α/β -hydrolase domain and a lid domain, which confers the substrate specificity and is reminiscent of feruloyl esterases.*’

5. The last line should be rephrased to indicate that these experiments are included in the paper i.e. ‘The structure has been used to inform....’, ‘which will serve as a basis for engineering MHETase for enzymatic plastic degradation’ or similar.

Assuming that ‘last line’ refers to the abstract, we now emphasize that the experiments are included in the paper by the addition of ‘reported here’. It now reads: *‘In the light of structure-based mapping of the active site, activity assays, mutagenesis studies and a first structure-guided alteration of substrate specificity towards bis(hydroxyethyl) terephthalate (BHET) reported here, we anticipate MHETase to be a valuable resource to further advance enzymatic plastic degradation.’*

6. Intro Line 21 ‘..moiety, rendering it..’

Corrected.

7. Page 3 line 1 ‘they are also’

Corrected

8. Page 3 line 2 – please define/expand ‘PET layers’

In the initial publication by Yoshida et al., *I. sakaiensis* was cultivated on thin PET films with low-crystallinity. This information was now added to the manuscript which reads: *Recently, the bacterial strain Ideonella sakaiensis was discovered and shown to grow on low-crystallinity PET films.’* and *‘Compared to known PET-degrading esterases, the PETase from I. sakaiensis shows higher activity at ambient temperature and on highly crystalline PET’*

9. Page 3 line 13 should read ‘mode of catalysis’ or ‘residues responsible for catalysis’ etc. and then ‘even permitted the enhancement of..’

‘Recent crystal structures of PETase bound to ligands confirmed the predicted α/β -hydrolase fold, elucidated substrate binding, catalysis and even allowed to slightly enhance catalytic properties or alter substrate specificity’ is now changed to *‘Recent crystal structures of PETase bound to ligands confirmed the predicted α/β -hydrolase fold, elucidated substrate binding, mode of catalysis and even permitted the enhancement of catalytic properties or alteration of substrate specificity.’*

10. Page 3 line 16 – what are ‘clear’ homologs – state limits of sequence ID

Since there is highly similar sequence data available (e.g. an entry of a Sulfitobacter tannase; WP_067951689.1) we have deleted the entire passage *‘Despite the absence of well-conserved homologs,’*.

11. Page 3 line 21 the enzyme does not ‘harbor’ a scaffold – please rephrase

‘Harbor’ is now replaced by ‘possess’

12, Line 22 – ‘..and expand substrate..’

The sentence ‘*MHETase, however, likely harbors a scaffold unprecedented for plastic-degrading enzymes, which may be exploited to improve catalysis and substrate specificity and thus significantly advance enzymatic plastic polymer degradation.*’ was rephrased to ‘*MHETase, however, is likely to possess a scaffold unprecedented for plastic-degrading enzymes. This may be exploited in order to improve catalysis and to expand substrate specificity and thus significantly advance enzymatic plastic polymer degradation.*’

13. Page 4 line 6 – state which structure (PDB code) and sequence ID

We agree that the PDB code should be given here. The reference ‘*supplementary methods*’ was now corrected to ‘*materials and methods*’. Therein, the passage ‘*an unpublished A. oryzae feruloyl esterase, AoFaeB-2; 26% identity to MHETase with a query coverage of 87%*’ was added where the MR procedure is called.

14. Line 7 - the lid domain is not ‘interspersed’ please rephrase

The entire sentence ‘*The overall structure of the 65 kDa MHETase resembles that of feruloyl esterases, with a lid domain interspersed between β -strand 7 and α -helix 16 (residues 254 – 468) of a hydrolase fold (Fig. 1a, Supplementary Figs. S1, S2a).*’ was rephrased to ‘*The overall domain architecture of the 65 kDa MHETase resembles that of feruloyl esterases, with a lid domain inserted between β -strand 7 and α -helix 15 of the α/β -hydrolase fold (Fig. 1b, Supplementary Fig. S1).*’ Residue numbers can be extracted from the alignment in Supplementary Fig. S1.

15. Line 9-10 – the presence of Ca is confirmed by X-ray FS, but not the ‘site’; please rephrase

We agree and rephrased ‘*As previously observed for feruloyl esterases, MHETase has a conserved calcium-binding site which was confirmed by X-ray fluorescence spectroscopy (Fig. 1a, Supplementary Fig. S2b).*’ to ‘*As previously observed for feruloyl esterases, the presence of a structural calcium-binding site was confirmed by X-ray fluorescence spectroscopy for MHETase (Fig. 1b, Supplementary Fig. S2b).*’

16. Line 12 ‘backbone amide nitrogens of G132..’

Corrected to ‘*the backbone amide nitrogen atoms of G132*’.

17. Page 5 line 13 'substrate specificity'

Corrected.

18. Line 17 remove 'specific'

Done!

19. Page 6 line 15 – 'very tightly' please give comparative values of K_m presumably

To be more precise, '*with a K_m of 7.3 μM* ' was added.

20. Page 7 line 6 rephrase 'addressed'

Here, '*addressed*' is now replaced by '*contacted*'

21. Rephrase line 8 'Contrasting MHETase..' – do you mean: 'By contrast, in MHETase, ' (?)

We admit that this may need clarification and certainly a link to the preceding sentence. We rephrased to '*Thus, the contribution of the LptE lid domain to substrate binding is much reduced, compared to the situation in MHETase.*'.

22. Line 14-15 'or pNP respectively'

Done!

23. Page 9 line 1 – A comparison of ...(what?) of potential ligands..'

We agree and included '*MHETase*' twice and rephrased as follows: '*A comparison of potential MHETase ligands with limited variations by DSF confirms, which functional groups are recognized by the MHETase binding site (for ligand quality control see Supplementary Fig. S7a-f).*'

24. Line 14 and 18 – check notation of K_m – two different uses here

Corrected

25. Line 21 'also the likely cause'

Corrected

26. Page 12 Line 12 – the evolution question posed here does not seem to belong to the discussion as it stands

We still consider the question how such a specific activity in *I. sakaiensis* may have evolved important. Admittedly, this passage was placed out of the general context and we now introduce the discussion with this evolutionary aspect. The corresponding passage in the results section is now also placed differently (earlier in the text).

27. Line 25 – this suggestion seems to demand data – is the activity of MHETase with this substrate known? Otherwise ‘alternative polyesters’ is probably a better general statement to finish with.

We agree with the reviewer and have now detailed a future direction for potential applications of MHETase variants in PEF degradation. We are aware that there is no data so far on PEF degradation available but think that this is a direction worthwhile to follow. We have changed and extended the entire text passage to: *‘With the structures of MHETase available, our detailed insights into its mechanism and in particular the generation of a BHETase with altered substrate specificity, it will be now possible to rationally create even more efficient MHETase variants cleaving other partial degradation products from related polymers. Replacing terephthalic acid in PET by thiophen-, furan- or pyridine-dicarboxylic acid has long been described²⁹. Exchanging the carboxylic esters by sulfonic esters in polymers is also possible³⁰. Polyesters of 2,5-furandicarboxylate with ethylene glycol (PEF) or other alcohols are suitable to replace PET in bottles³¹. This new plastic PEF can be degraded by PETase¹⁹. The product hydroxyethyl-2,5-furandicarboxylate is similar enough to MHET to envision structure guided mutagenesis of MHETase to evolve a ‘MHEFase’ for the full cycle from renewable carbohydrates to PEF and back to polymer building blocks by green chemistry.’*

28. Page 16 – Line 19 – state what this structure is (6G21)

It is now explained in detail in the materials and methods section. *‘an unpublished A. oryzae feruloyl esterase, AoFaeB-2; 26% identities to MHETase with a query coverage of 87%.’*

29. References –there are some inconsistencies in page numbering etc. e.g. 26, 37

We have re-formatted the references.

30. The structures are not deposited- the validation reports submitted clearly state that those reports are not to be submitted to journals - PDB codes must be included in a revised submission.

All four structures have been deposited at the PDB, the information was added to the data availability statement. *'Structure coordinates and diffraction data were deposited with the Protein Data Bank (<http://www.pdb.org>) under accession codes 6QG9 (MHETase), 6QGA (MHETase MHETA), 6QGB (MHETase BA), 6QGC (PETase).'*

31. Nature Comms asks that 'To facilitate assessment of the quality of the structural data, a stereo image of a portion of the electron density map' [should be provided] – it is suggested that this is done for the density figure in the SI.

All refined 2Fo-Fc electron densities of the four structures are now shown as stereo images in Supplementary Figure S2c-f.

32. Please check significant figures for the Crystallography Data Table – resolution, R-factors (3 dec places or 22.1% etc.), B factors (integer values)

All figures in the Crystallography Data Table were refined regarding consistency. As XDS does not report the third decimal, the R-factors in the refinement section were truncated, accordingly. The B-factors are now formatted as integer values.

33. Please provide separate B-factors for solvent and ligands

Done

Reviewer #3

This manuscript describes very extensively the structural investigation of *I. sakaiensis* MHETase which degrades in combination with *I. sakaiensis* PETase the synthetic polymer polyethylene terephthalate. Based on structural data, MHETase mutants were generated showing activity on BHET which is totally missing in wt MHETase. However, essential data are missing showing the effect of generated mutants in combination with PETase. How much BHET is formed by PETase and how significant is the effect of BHET activity for total PET degradation? Additionally, the section Materials and Methods has to be completely revised since important information is missing and several mistakes are in the text (see some comments below).

We thank the reviewer for commenting on our manuscript. We are also grateful for valuable suggestions regarding further experiments as well as pointing at mistakes in the 'Materials and Methods' section.

Major comments

Introduction, line 10: it is stated in line 6, that so far no enzyme has been found that fully penetrates and degrades a thick layer of highly crystalline PET in a cost-effective manner. What is the performance of the two enzymes, PETase and MHETase, compared to the already known and published enzymes? How much PET is degraded and in which time? What's the crystallinity of the PET used in the studies?

As the reviewer notes, an important challenge for the biotechnological application will be the improvement of the PETase/MHETase pair to degrade PET faster and at technically usable conditions. In contrast to that, this work aimed at identifying the structural prerequisites for MHET hydrolysis which we then, as a proof of concept, used to expand the activity to related building blocks like BHET.

However, we performed preliminary experiments on PET nano particles which indicate that addition of MHETase, in fact, increases the degradation rate of PET by PETase (i.e. the amount of produced TPA + MHET + BHET per PETase). It is also clear from these tests that PETase concentration, PETase/MHETase ratio and buffer composition play a role. More technologically relevant factors are likely to have an effect (enzyme variants, particle size, PET crystallinity, temperature). It is not scope of this paper to evaluate the influence of these parameters or even optimize them.

Page 3, line 16: assignment to the tannase family and sequence homology to the feruloyl esterases from *A.oryzae* should be confirmed by testing MHETase with selected substrates of this enzyme family.

We have now tested different substrates of the tannase family with MHETase and BHETase variants. Based on the data, we have added a text passage to the end of the results section and Figure S10 to the supplement. The text passage now reads: '*Lastly, wild-type MHETase and variants S416A F424N and R411A S419G F424N, which have BHETase activity, were also examined for activity towards coumaric acid methyl ester (Coum-ME), caffeic acid methyl ester (Caff-ME), chlorogenic acid (Chlorogen) and p-hydroxy benzoic acid methyl ester (pHB-ME), substrates for feruloyl and chlorogenate esterases. UV-Vis spectra using 10 μ M of the ester and the free acid were measured and $\Delta\epsilon$ calculated (Fig. S11). Hydrolysis was measured as for MpNPT, but with 10-35 nM enzyme, 100 μ M substrate and at 335 nm (Coum-ME, $\Delta\epsilon = -6100 \text{ M}^{-1}\text{cm}^{-1}$), 350 nm (Caff-ME, $\Delta\epsilon = -5700 \text{ M}^{-1}\text{cm}^{-1}$), 350 nm (Chlorogen, $\Delta\epsilon = -7400 \text{ M}^{-1}\text{cm}^{-1}$) and 280 nm (pHB-ME, $\Delta\epsilon = -3900 \text{ M}^{-1}\text{cm}^{-1}$). No activity above background (no enzyme) could be detected.'*

Page 3, line 2: what are the homologies for the hydrolase domains?

All relevant homologies were added to the respective text passages

Page 4, line 6: how high is the homology between MHETase and *A.oryzae* feruloyl esterase?

We have now included an overall comparison of MHETase and FaeB in the main text. '*The overall structures of MHETase and FaeB are structurally similar (2.04 Å RMSD for 421 / 559 residues aligned) despite a relatively low number of amino acid identities (27.5%).'*

Page 4, line 9: what is the function of the calcium ion?

After rephrasing this sentence as suggested by reviewer #2, we have now added 'structural' to the sentence as follows: '*As previously observed for feruloyl esterases, the presence of a structural calcium-binding site was confirmed by X-ray fluorescence spectroscopy for MHETase (Fig. 1b, Supplementary Fig. S2b).'*

Figure 1: is quite overloaded and repetitive, select and display most important information

The figure was now re-built entirely – also based on comments from reviewer #2

Page 5, line 22: the term “apo-MHETase” is quite confusing since apo-enzymes are defined as enzymes lacking a cofactor. In this manuscript, the term is used for description of the free enzyme in contrast to the enzyme-substrate-complex?

All instances of ‘apo’ were replaced by ‘ligand-free’.

Page 9, line 24: what do you mean with “comparably low activity of MHETase”? With which enzymes and activities do you compare the MHETase activities?

We have rewritten and rearranged this entire paragraph since we do not directly compare activities of MHETase with other enzymes here. The improved text passage addressing the induced-fit mechanism reads: ‘*Lastly, unlike PETase, MHETase binds to its substrate very tightly with a K_m of $7.3 \mu M^{17}$. A comparison of the active-site molecular surfaces of LptE, PETase and MHETase in their substrate-bound states illustrates a higher solvent accessibility of LptE and PETase, which is partially related to the induced-fit mechanism observed for MHETase and the number of residues contacting the respective substrate (Fig. 2c; d Supplementary Figure 5 a-d).*’ Another passage addresses product inhibition as follows: ‘*The high affinity for compounds with a benzoate substructure is expected to lead to product inhibition by formed TPA when higher concentrations of MHET are hydrolyzed in vitro. This is demonstrated by the decreasing reaction rate for MHET hydrolysis over time (Supplementary Figure 9a). The effect of product inhibition is likely less pronounced in the natural environment where formed TPA is metabolized by the bacterium.*’ Taken together, we have now two text passages which are supposed to unfold ways of improving MHETase. The sentence containing the phrase ‘comparably low activity’ has been removed.

Page 11, line 11: it is not explained why the missing activity on BHET is of importance; how much BHET is released in the course of PET degradation by the PETase? Is BHET the main product?

The missing activity of the MHETase wildtype towards BHET is not of importance as the PETase produces almost exclusively MHET and TPA (Fig. 2B, D in Yoshida *et al.*). Therefore, it is not required in nature to hydrolyze BHET by MHETase and thus, MHETase is fully specialized in binding MHET. To include this information, the following passage was added to the Discussion: ‘*This specialization for the natural substrate MHET also explains the very low activity of MHETase towards the intermediate BHET in the wild-type enzyme. In the natural environment, missing activity towards BHET is not critical as the upstream enzyme, PETase, already hydrolyzes BHET to MHET.*’ However, for biotechnological applications, it could be valuable to use a mutant MHETase with broader substrate spectrum.

Page 12, line 1: structural investigation of MHETase is a “key step in understanding the full process of microbial PET degradation by *I. sakaiensis*” but cannot be generalized.

We now refer to the microbial PET degradation in *I. sakaiensis* by rephrasing to ‘*The first elucidation of the structure of MHETase represents a key step in understanding the process of microbial PET degradation in I. sakaiensis.*’

Page 12, line 6: it is stated that MHETase variants can degrade several PETase products down to monomers; what do you mean with “several PETase products” except of MHET and BHET?

We agree and have amended the sentence to ‘*We were thus successful in generating a MHETase variant, which hydrolyzes the PETase products MHET and BHET down to the very building blocks, which are required for a sustainable re-synthesis of the polyethylene terephthalate polymer.*’

Page 12, line 17: also for the evolution of MHETase from tannase it would be essential to verify MHETase activity towards tannase substrates.

We have now tested different substrates of the tannase family with MHETase and BHETase variants. Based on the data, we have added a text passage to the end of the results section and Figure S10 to the supplement. The text passage now reads: *Lastly, wild-type MHETase and variants S416A F424N and R411A S419G F424N, which have BHETase activity, were also examined for activity towards coumaric acid methyl ester (Coum-ME), caffeic acid methyl ester (Caff-ME), chlorogenic acid (Chlorogen) and p-hydroxy benzoic acid methyl ester (pHB-ME), substrates for feruloyl and chlorogenate esterases. UV-Vis spectra using 10 μ M of the ester and the free acid were measured and $\Delta\epsilon$ calculated (Fig. S11). Hydrolysis was measured as for MpNPT, but with 10-35 nM enzyme, 100 μ M substrate and at 335 nm (Coum-ME, $\Delta\epsilon = -6100 \text{ M}^{-1}\text{cm}^{-1}$), 350 nm (Caff-ME, $\Delta\epsilon = -5700 \text{ M}^{-1}\text{cm}^{-1}$), 350 nm (Chlorogen, $\Delta\epsilon = -7400 \text{ M}^{-1}\text{cm}^{-1}$) and 280 nm (pHB-ME, $\Delta\epsilon = -3900 \text{ M}^{-1}\text{cm}^{-1}$). No activity above background (no enzyme) could be detected.*’

The result shows that MHETase has diverged substantially from its relatives and we state in the discussion: ‘*Typical tannase substrates like hydroxy cinnamates and hydroxy benzoates, are not converted by MHETase (as shown already¹⁰) nor by the MHETase variants with BHETase activity which we engineered and present here.*’

Page 12, line 23: which polymers do you have in mind that contains heteroaromatics or sulfonic acid groups?

We have now rephrased and detailed this text passage as follows: *‘With the structures of MHETase available, our detailed insights into its mechanism and in particular the generation of a BHETase with altered substrate specificity, it will be now possible to rationally create even more efficient MHETase variants cleaving other partial degradation products from related polymers. Replacing terephthalic acid in PET by thiophen-, furan- or pyridine-dicarboxylic acid has long been described²⁹. Exchanging the carboxylic esters by sulfonic esters in polymers is also possible³⁰. Polyesters of 2,5-furandicarboxylate with ethylene glycol (PEF) or other alcohols are suitable to replace PET in bottles³¹. This new plastic PEF can be degraded by PETase¹⁹. The product hydroxyethyl-2,5-furandicarboxylate is similar enough to MHET to envision structure guided mutagenesis of MHETase to evolve a ‘MHEFase’ for the full cycle from renewable carbohydrates to PEF and back to polymer building blocks by green chemistry.’*

Page 13, line 10: what is the degree of crystallinity? Since it is described in the introduction that *I. sakaiensis* can degrade high crystalline PET this information is important.

The crystallinity of the starting material for BHET and MHET synthesis is not relevant for our study as stated above.

Page 13 and 14: chemical and structural analysis of synthesized molecules as well as determination of purities are fully missing.

We have now added Supplementary Figure 7 with the respective analyses.

Page 16: here, the crystallization and analysis of PETase is described, but in the text it is described that this is another publication?

We agree that the structure solution of ligand-free PETase is missing in the main text. The structure matches to prior ligand-free PETase structures and is therefore not discussed further and only shown for comparison of the α/β hydrolase fold in Fig. S2b. We now added *‘The structure of PETase was solved by molecular replacement (PHASER) employing the structural coordinates of *T. fusca* cutinase TfCut2 (PDB entry 4CG1^{11,23}).’* to the main text.

Page 18, line 8: what was the reaction mixture? Provide details regarding enzyme concentration, reaction volume, substrate concentration, reaction conditions etc.

We have now stated the volume of the reaction (100 μ L) mixture in the Methods section. The other requested details were already mentioned in the following paragraph.

Minor comments

Page 5, line 6: give the full name for *L. plantarum* since it is mentioned for the first time in the manuscript

Done

Page 7, line 10: provide the full name for HEMT and pNP

Full names are now provided.

REVIEWERS' COMMENTS:

Reviewer #1 (Remarks to the Author):

All of my comments have been implemented.

Reviewer #3 (Remarks to the Author):

In their revised manuscript, the authors have carefully addressed all the Reviewers concerns and hence the paper is now acceptable for publication